# Critical Output Torque of a GHz CNT-Based Rotation Transmission System Via Axial Interface Friction at Low Temperature

**DOI:** 10.3390/ijms20163851

**Published:** 2019-08-07

**Authors:** Puwei Wu, Jiao Shi, Jinbao Wang, Jianhu Shen, Kun Cai

**Affiliations:** 1College of Water Resources and Architectural Engineering, Northwest A&F University, Yangling 712100, China; 2State Key Laboratory of Structural Analysis for Industrial Equipment, Dalian University of Technology, Dalian 116024, China; 3School of Port and Transportation Engineering, Zhejiang Ocean University, Zhoushan 316022, China; 4Centre for Innovative Structures and Materials, RMIT University, Melbourne 3083, Australia

**Keywords:** rotation transmission system, interface friction, nanotube, torque moment, molecular dynamics

## Abstract

It was discovered that a sudden jump of the output torque moment from a rotation transmission nanosystem made from carbon nanotubes (CNTs) occurred when decreasing the system temperature. In the nanosystem from coaxial-layout CNTs, the motor with specified rotational frequency (*ω*_M_) can drive the inner tube (rotor) to rotate in the outer tubes. When the axial gap between the motor and the rotor was fixed, the friction between their neighbor edges was stronger at a lower temperature. Especially at temperatures below 100 K, the friction-induced driving torque increases with *ω*_M_. When the rotor was subjected to an external resistant torque moment (*M*_r_), it could not rotate opposite to the motor even if it deformed heavily. Combining molecular dynamics simulations with the bi-sectioning algorithm, the critical value of *M*_r_ was obtained. Under the critical torque moment, the rotor stopped rotating. Accordingly, a transmission nanosystem can be designed to provide a strong torque moment via interface friction at low temperature.

## 1. Introduction

Carbon materials with wide application have played an important role in material interface projects for over half a century [1]. In particular, since specific carbon nanotubes (CNTs) and graphene can be fabricated [2,3,4,5,6,7], low dimensional materials start to enter into the public eye due to their excellent physical properties. Their coming-out also inspires wide interest in developing new low dimensional materials to meet the requirement of the materials with specified features [2,8,9,10,11].

Due to particular electron configurations, carbon materials behave extremely high strength via the 2sp2- and/or 2sp3 covalent bonds. For example, the sp2-sp2 carbon materials, e.g., fullerene, CNT, and graphene, have large strength and modulus [12,13,14,15,16,17]. On the other hand, each carbon atom also has an antibonding electron, which introduces two particular properties, i.e., excellent electrical conductivity [18,19] and superlubrication [4,7,20,21] between two neighbor tubes or sheets. Owing to both extremely high in-plane/shell strength and modulus and extremely low inter-plane/shell friction, CNTs and graphene are popular in developing nanodevices, e.g., oscillator [22,23,24,25,26,27,28], nanomotor [29,30,31], nanobearing [32], nanoresonator [33,34,35,36].

In studying the inter-tube friction, generally, multi-walled CNTs were used in experiments or simulations [7,22]. In the mechanics point of view, friction is caused by asynchronous movement among nonbonding atoms. Hence, we conclude that the two end-to-end concentric tubes must have friction when their relative sliding happens. According to this conclusion, Cai et al. [37] proposed a concept of rotation transmission system from concentric nanotubes, which looks similar to Figure 4a. Further studies [38] demonstrate that the interface friction between the edges of two tubes depends on the difference of their rotational speeds. Chirality of tube and temperature were also considered. All the works were focused on the transmission effect, which is measured by the rotation transmission ratio, i.e., the rate between the rotational frequencies of the rotor and the motor.

In this study, we researched the effect of interface friction on the rotational transmission effect. According to dynamics, a tube is driven to rotate when it is not in moment equilibrium about the rotational axis. Hence, the friction-induced torque moment from the motor to the rotor is suitable for describing the interface effect. To avoid the influence of friction from the stators, an active resistant moment was applied on the rotor when measuring the torque moment from the high-speed rotating motor.

## 2. Results and Discussion

### 2.1. Rotation Transmission of the Zigzag Model

We first measured the value of rotation transmission ratio, i.e., *R*_Tran_, as shown in Figure 1, of the zigzag nanosystem with *M*_r_ = 0 eV at different conditions. In this case, the resistant torque moment was only from the two stators. The rotor has no axial translation because the axial translation freedom of its right edge was fixed. The results are listed in Table 1. When the temperature of the system was higher than 100 K, *R*_Tran_ decreased with increasing *ω*_M_. For example, at 500 K, it decreased from ~0.96 at *ω*_M_ = 50 GHz to ~0.11 at *ω*_M_ = 200 GHz. Hence, both *ω*_M_ and temperature influenced the rotational frequency of the rotor. The reason is that the interaction at the interface edges depends on the two main factors. For example, at high temperature, the tubes have thermal expansion both in the axial and the radial directions. Considering both axial thermal expansion (thermal expansion coefficient is negative when T < 300 K, otherwise positive when T > 300 K [39]) and the existence of edge barriers [40], the distance between the interface edges of the motor and the rotor was less than 0.5 nm. When the distance was very close to 0.34 nm, their interaction became very low according to the 12-6 type Lennard-Jones potential in the AIREBO potential for describing the nonbonding interaction between atoms. When the distance is between 0.2 nm and 0.34 nm, strong repulsion will increase the interface friction, and further leads to larger rotational frequency of the rotor, i.e., *ω*_R_. However, as mentioned above, the friction between the rotor and the stators increased with *ω*_R_. Therefore, the lower value of *ω*_M_ led to a higher value of *R*_Tran_ when the temperature was not lower than 100 K according to the results listed in Table 1.

However, *R*_Tran_ increased with the input rotational frequency on the motor when the temperature was lower than 100 K. For example, at 50 K, it increased from ~0.54 at *ω*_M_ = 50 GHz to ~0.80 at *ω*_M_ = 200 GHz. The reason is that, at a lower temperature, thermal vibration of the edge atoms introduced weaker axial collision between the motor and the rotor. Meanwhile, the friction from the two stators did not increase with *ω* as high as that at a higher temperature, i.e., *M*_M_(*ω*) − (*M*_S1_ + *M*_S2_) grew with the increasing of *ω*. Hence, *R*_Tran_ increased with *ω* at 50 K or a lower temperature. Briefly, as the temperature decreased, the temperature effect on the inter-tube friction between the rotor and the stators became weaker, and the rotation transmission efficiency was mainly determined by the rotor’s rotation when the right edge of the rotor was confined in its axial movement.

Secondly, we searched the values of *M*_r_^cr^ of the zigzag system at different conditions with the bi-section algorithm and listed the results in Table 2. It states that the torque moment transferred via the interface was lower at a higher temperature when *ω*_M_ was constant. In fact, two special phenomena attracted our attention. One was the sudden jump of the critical value of the applied resistant torque moment for a system with specified *ω*_M_ in a cooling/heating process. For instance, when *ω*_M_ = 150 GHz, the critical value was ~175 eV at 50 K. If the temperature increased to 100 K or higher, the critical value became less than 1.0 eV. Similarly, when *ω*_M_ = 200 GHz, the critical value varied from ~130 eV at 100 K to ~0.5 eV at 300 K. The reason is that the motor provided strong friction with the motor at their end interface at low temperature because the friction from the stators can be neglected at such low temperature and very low rotational speed of rotor according to Equation (10). It implies that a strong power can be output by the rotor via the interface friction with the high speed rotating motor at low temperature. It also suggests an easy way to control the output just changing the temperature of the system.

The other is the rotor buckled at 8 K when *ω*_M_ was higher than 100 GHz and the resistant torque moment was too large (Figure 2c). It means that the transmission efficiency became higher at a lower temperature for the same system. From the snapshots shown in Figure 2, we can find that the rotor tube did not deform obviously when *M*_r_ = −175 eV at 50 K. The two tubes rotated oppositely (Appendix A). The motor rotated around within 7 ps but the rotor needed about 3700 ps for a round of rotation, which is much lower than that of the motor. However, at 8 K, if *M*_r_ = −205 eV, the rotor deformed obviously after about 50 ps of rotation. Its loading end was buckled and enclosed at 56 ps. Later the whole tube was buckled (Appendix A). It was not caused by the friction from the stators. This can be verified by the state of the system at 8 K with *ω*_M_ = 200 GHz, i.e., the value of *R*_Tran_ >0.5 when *M*_r_ = -217.13 eV. It means that the rotor has over 100 GHz of the rotational frequency with the same direction as that of the motor when 217.13 eV of resistant torque was applied on its free end. If the resistant torque moment increased slightly, e.g., from 217.13 eV to 217.23 eV, the rotor tube buckled rapidly.

### 2.2. Rotation Transmission of the Armchair Model

When using armchair CNTs to form the system, the rotor rotated almost synchronously with the motor at 8 K if there was no external resistant torque moment on the rotor. When *ω*_M_ was larger than 100 GHz, the value of *R*_Tran_ decreased with the increasing temperature (Table 3). Hence, the temperature effect on the armchair model was identical to that on the zigzag model. Compared with the zigzag model, the armchair model had better rotation transmission efficiency, but poor stability of the rotor’s rotation at 500 K or when *ω*_M_ was lower than 100 GHz. This is because the potential barriers on the armchair surface were aligned with generatrix. At high temperature, thermal vibration increased the friction from the stator but decreased the friction from the motor via their end interface. Hence, rotation transmission was not stable, which could be verified from the fluctuation of the curves in Figure 3.

If an external resistant torque moment was applied on the rotor, the rotor rotated slower or even rotated in the direction opposite to the motor. Using the bi-section method, the critical value of the resistant torque moment was captured for each case. For example, at 8 K, the critical value was ~3.0 eV when *ω*_M_ = 50 GHz. At 50 K, the critical value was ~0.75 eV when *ω*_M_ = 100 GHz. According to the results listed in Table 4, the critical value decreased with the increasing temperature when the system had a fixed value of *ω*_M_. For instance, when *ω*_M_ = 50 GHz, *M*_r_^cr^ is ~ 0.35 eV at 50 K and became 0.075 eV at 500 K.

In the armchair model, we also found that the value of the resistant torque moment had a sudden jump with the changing temperature. For example, when *ω*_M_ = 150 GHz, *M*_r_^cr^ is 1.0 eV at 100 K and jumped up to 130 eV at 50 K. for the same system, the rotor buckled when *M*_r_ = 260 eV before reversing its rotational direction. At temperature below 100 K, the rotor could not be stopped by the external resistant torque moment when the motor had a high rotational frequency, e.g., 150 GHz or higher. Hence, cooling the system will lead to stronger torque transferred from the motor to the rotor via their end interface. This conclusion is identical to that for the zigzag model, as well.

## 3. Model and Methods

### 3.1. Model

In this study, both the zigzag and armchair models of the rotation transmission system (Figure 4) were tested using the molecular dynamics approach. At finite temperature, thermal vibration of atoms on tubes influences the interaction between two neighbor tubes. According to previous studies [41], the input rotational frequency of the motor, i.e., *ω*_M_, also influences the transmission effect. Hence, 5 temperatures between 8 K and 500 K together with 4 input rotation between 50 GHz and 200 GHz were considered in the tests.

Driven by the motor, the rotor rotates in the same direction but its rotational frequency may be different from the motor [37]. In general, weak torque moment may be transferred via the hydrogenated edges and is easily balanced by the friction-induced moment from the stators. If they are made from the same tube, the rotor has at most the same rotational frequency as the motor [42]. However, it was difficult to measure the output torque moment on the rotor via its stable rotational frequency (SRF). To solve this difficulty, we provided an active resistant torque moment on the right (orange) edge of the rotor in the nano-bearing. The resistant torque moment had an opposite direction to the input rotational direction, and the rotor had 3 possible states. First, the rotor rotated slower but had the same direction as the motor when the magnitude of *M*_r_ was very small. Second, if *M*_r_ was too large, the rotor had an opposite rotational direction to the motor. Finally, when *M*_r_ was in an interval, the rotor’s rotational frequency, i.e., *ω*_R_, was far less than *ω*_M_. Hence, we could find the lower boundary of the interval using the bi-section algorithm. This value is called the critical resistant torque moment and labeled as “*M*_r_^cr^”. The major task of this study is to measure the quantity of *M*_r_^cr^ on the rotor under different conditions.

### 3.2. Methodology

#### 3.2.1. Mechanism of Rotation Transmission

In the transmission model shown in Figure 4a, the interaction between the edges of the motor and the rotor was very strong when *d* = 0.5 nm. As *d* is greater than the equilibrium distance of 2 nonbonding carbon atoms, i.e., 0.34 nm, hence, the motor provided an attraction onto the rotor via the edge carbon atoms. When the motor is rotating, the attractive force from each carbon atom on the motor will have a circumferential component, which is the actuating force to drive the rotor to rotate. The resultant effect of the circumferential forces from the edge atoms is described by a torque moment and labeled as “*M*_M_”. When the rotor is rotating with the motor along the same direction, friction-induced resistant moments from the two stators are increasing. If there is no active moment applied on the right edge of the rotor, the rotational frequency of the rotor can be expressed as
(1)$$\omega \left(t\right)=\frac{1}{2\pi }\int_{0}^{t}\frac{M_{M}\left(\tau \right)-M_{S1}\left(\tau \right)-M_{S2}\left(\tau \right)}{J_{z}}d\tau$$
where *J*_z_ is the moment of inertia of the rotor about Z-axis. When the driving moment is balanced by the resistant moment, the rotor has a stable rotational frequency (SRF), i.e.,
(2)$$\omega_{R}=\omega \left(t^{*}\right),\;\mathrm{if}\;M_{M}\left(t^{*}\right)-M_{S1}\left(t^{*}\right)-M_{S2}\left(t^{*}\right)=0$$

Now, an active moment is applied on the right edge of the rotor, similarly, the SRF of the rotor can be expressed as
(3)$$\omega_{R}=\omega \left(t\geq t^{*}\right)=\int_{0}^{t}\frac{M_{M}\left(\tau \right)-M_{S1}\left(\tau \right)-M_{S2}\left(\tau \right)-M_{r}\left(\tau \right)}{2\pi \cdot J_{z}}d\tau$$

One possible case is that the rotor may rotate opposite to the motor once the active resistant moment is too large. In this case, the directions of *M*_S1_ and *M*_S2_ have the same direction as *M*_M_, and the rotational frequency of the rotor is negative, i.e.,
(4)$$\omega_{R}=\omega \left(t\geq t^{*}\right)=\int_{0}^{t}\frac{M_{M}\left(\tau \right)+M_{S1}\left(\tau \right)+M_{S2}\left(\tau \right)-M_{r}\left(\tau \right)}{2\pi \cdot J_{z}}d\tau <0$$

Hence, *M*_S1_ and *M*_S2_ will be much less than *M*_M_ or *M*_r_ when the rotor has no rotation on time average. And we conclude that the *M*_M_ is approximately equal to *M*_r_ when the absolute value of *ω*_R_ is much larger than *ω*_M_. According to this conclusion, we build a model to test the transmission torque via the interface between the motor and the rotor.

#### 3.2.2. Bi-Section Algorithm for Finding M_r_^cr^

The value of *M*_r_^cr^ can be found when the rotor’s rotational frequency is far less than that of the motor. For simplicity, the rotation transmission ratio (RTR) of the system is defined as
(5)$$R_{\mathrm{Tran}}=\frac{\omega_{R}}{\omega_{M}}$$

Respectively, the value of *M*_r_^cr^ reads
(6)$$M_{r}^{\mathrm{cr}}=M_{r}(\left|\omega_{R}/\omega_{M}\right|<<1)$$

To find the value of *M*_r_^cr^, the bi-section algorithm is adopted. In this algorithm, an initial interval of *M*_r_, e.g., [*a*_0_, *b*_0_], should be provided. When considering anticlockwise rotation implies a positive value of *ω*_R_ and *ω*_M_, *a*_0_ and *b*_0_ should satisfy the following inequation,
(7)$$\omega_{R}(a_{0})\cdot \omega_{R}(b_{0})<0$$

The mid value of the interval is *c_i_*_+1_, i.e.,
(8)$$c_{i+1}=(a_{i}+b_{i})/2$$

Briefly, the flowchart of the bi-section algorithm for the present problem is as following,

Step 1 Let *i* = 0; find [*a_i_*, *b_i_*] satisfies Equation (7);

Step 2 Calculate *c_i+_*_1_ using Equation (8), find the value of *ω*_R_(*c_i+_*_1_);

Step 3 If $\omega_{R}(a_{i})\times \omega_{R}(c_{i+1})<0$, let $a_{i+1}=a_{i},\;b_{i+1}=c_{i+1}$, and go to Step 4; else if $\omega_{R}(a_{i})\times \omega_{R}(c_{i+1})>0$, let $a_{i+1}=c_{i+1},\;b_{i+1}=b_{i}$, and go to Step 4; else $\omega_{R}(a_{i})\times \omega_{R}(c_{i+1})=0$, go to Step 5;

Step 4 If $\left|R_{\mathrm{Tran}}\right|<0.005$ or $\left(b_{i+1}-a_{i+1}\right)/\left(a_{i+1}+b_{i+1}\right)<0.001$, go to Step 5; otherwise, let *i* = *i* + 1, go to Step 2;

Step 5 Stop iteration, and record *c_i+_*_1_ as the value of *M*_r_^cr^.

#### 3.2.3. Molecular Dynamics Simulation Approach

Molecular dynamics approach is adopted to illustrate the dynamics response of the rotor when driven by both the motor and the resistant torque moment. In this study, molecular dynamics simulations were carried out on the open source code LAMMPS [43,44]. In each simulation, the interaction between carbon and/or hydrogen atoms was evaluated by AIREBO potential [45]. After building the model of the system, the components were reshaped by energy minimization. Further, the stators were fixed at their outer edges (Figure 4), some atoms on the motor and the rotor were fixed before starting 200 ps of thermal bath in the NVT ensemble with Nose-Hoover thermostat [46,47] controlling the temperature of the system. After that, an input rotation was applied on the motor and a resistant torque moment on the rotor when releasing their fixed atoms. Simultaneously, the significant data of the system was recorded for potential analysis. For time integral, the timestep was set to be 0.001 ps. No less than 10 ns of the dynamic response of the system was simulated in each case.

#### 3.2.4. Temperature Effect on Torque Transmission

In dynamics simulation, the temperature of the system was controlled by the Nose-Hoover algorithm, and the velocities with respect to thermal vibration of atoms were in the normal distribution. For the edge atoms on the interface between the motor and rotor, at a higher temperature, they have larger amplitude vibration along the radial direction. On time average, the attraction between the motor and the rotor becomes weaker. Hence, output torque via the interface will be lower for the same system. To evaluate the effect, transmission states at different temperatures were studied.

#### 3.2.5. Effect of Input Rotation on Torque Transmission

As mentioned earlier, the friction between the rotor and the stators increases with increasing of their relative sliding speed, i.e., *ω*_R_. Hence, the motor can also provide stronger friction on the rotor by the input of higher rotational frequency. This is another way to change the output torque moment via the interface for the same system.

When there was no active resistant torque moment, i.e., *M*_r_ = 0, on the rotor, which was in stable rotation, the resistant torque from the 2 stators could balance the active torque moment from the motor. Consider both torque moments as functions in terms of the rotational frequencies, their relationship yields
(9)$$M_{M}\left(\omega_{M}-\omega_{R}\right)=M_{S1}\left(\omega_{R}\right)+M_{S2}\left(\omega_{R}\right)$$

If the *ω*_M_ is twice of *ω*_R_, it means that the resultant resistant torque moment from the two stators is equal to that from the motor via the interface.

Once *M*_r_ ≠ 0, and *ω*_R_ = 0, according to Equations (3) and (4) we have *M*_S1_(*ω*_R_) = *M*_S2_(*ω*_R_) = 0, and
(10)$$M_{M}\left(\omega_{M}\right)=M_{r}$$

In this case, *M*_M_ is a dual quantity of *ω*_M_ with respect to the output power of the motor via the interface.

## 4. Conclusions

Using coaxial-layout carbon nanotubes (CNTs), we formed a rotation transmission system, which contained a motor with fixed rotational frequency (*ω*_M_) and a nano-bearing from double-walled CNTs. In the nano-bearing, the outer tubes were partly fixed as stators, and the inner tube acted as a rotor. Without applying external resistant torque moment (*M*_r_) on it, the rotor would be driven to rotate with the motor via friction at their end interface. Using molecular dynamics simulations together with the bi-sectioning algorithm, the critical value of *M*_r_ was obtained. Under the critical torque moment, the rotor’s rotational frequency was zero or much less than *ω*_M_. When searching the critical values, two important phenomena were discovered.

One is that the critical value of *M*_r_ has a sudden jump when varying temperatures. It increases more than 100 times when temperature decreased from 100 K to 50 K or to 8 K. This implies that one can adjust the output torque from the rotor by decreasing the system temperature. The other is that the rotor cannot rotate opposite to the motor even if it was buckled by strong external resistant torque moment. More importantly, the phenomena are independent on the chirality of the CNTs in the system.

According to the present study, one can design a transmission system to provide a strong torque moment via the interface between the neighbor ends of the motor and the rotor at low temperature.

## Figures and Tables

**Figure 1 ijms-20-03851-f001:**
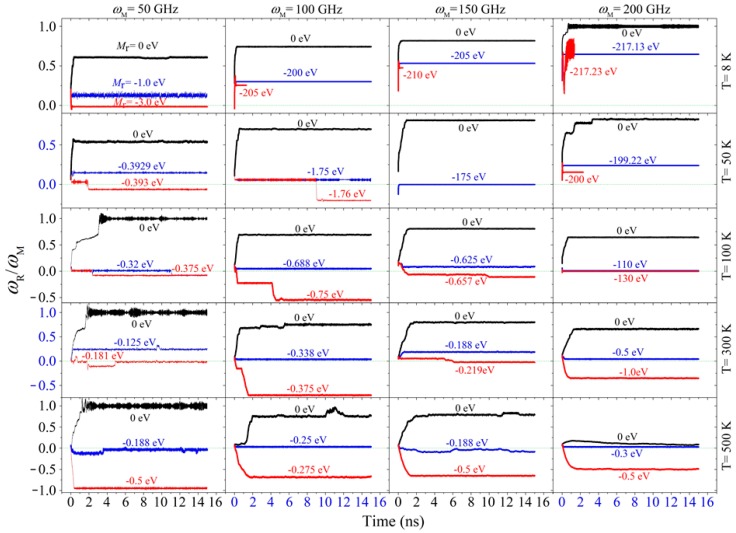
Historical curves of the rotation transmission ratio, i.e., *R*_Tran_ in Equation (5), of the zigzag model with Gap = 1.5 nm and subjected to *M*_r_ at different conditions. Note that the curves in each column are obtained at the same value of input rotation, e.g., “*ω*_M_ = 50 GHz” means that the curves in the left column are corresponding to the system with input rotational frequency of 50 GHz. Each row contains the curves obtained at the same temperature, e.g., “T = 8 K” says the curves in the top row are obtained at 8 K. Not all the results involved in the bi-section algorithm were shown here. Only three of them in each case were listed for showing the critical values.

**Figure 2 ijms-20-03851-f002:**
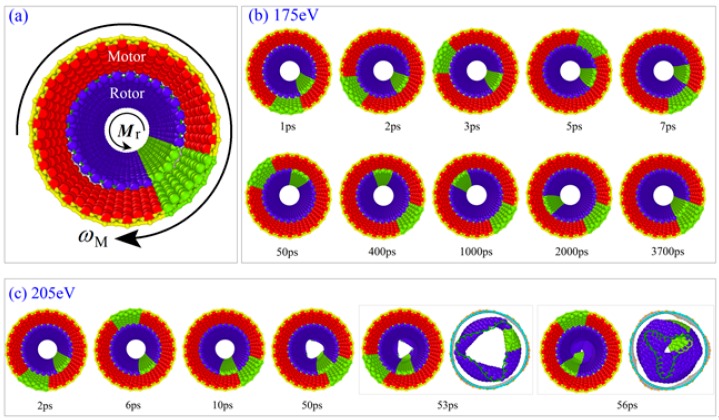
Relative rotation between motor and rotor in the zigzag model. (**a**) The perspective of the initial configuration after relaxation. (**b**) Snapshots of the system with *M*_r_ = 175 eV, *ω*_M_ = 150 GHz at 50 K. The rotor rotates a round after about 3700 ps. (**c**) Snapshots of the system with *M*_r_ = 205 eV, *ω*_M_ = 100 GHz at 8 K. The rotor has obvious buckling at 50 ps, the loading end of the rotor was enclosed after 56 ps.

**Figure 3 ijms-20-03851-f003:**
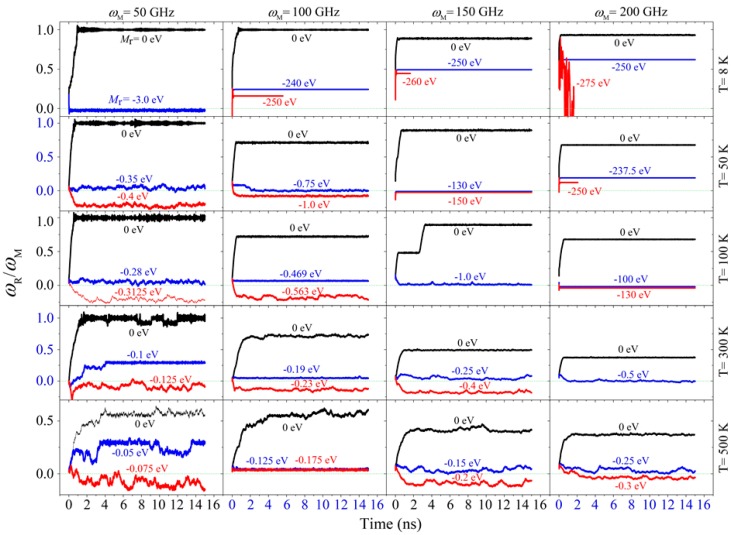
Historical curves of the rotation transmission ratio, i.e., *R*_Tran_, of the armchair model at different conditions. Not all the results involved in the bi-section algorithm were shown here. Only three of them in each case were listed for showing the critical values.

**Figure 4 ijms-20-03851-f004:**
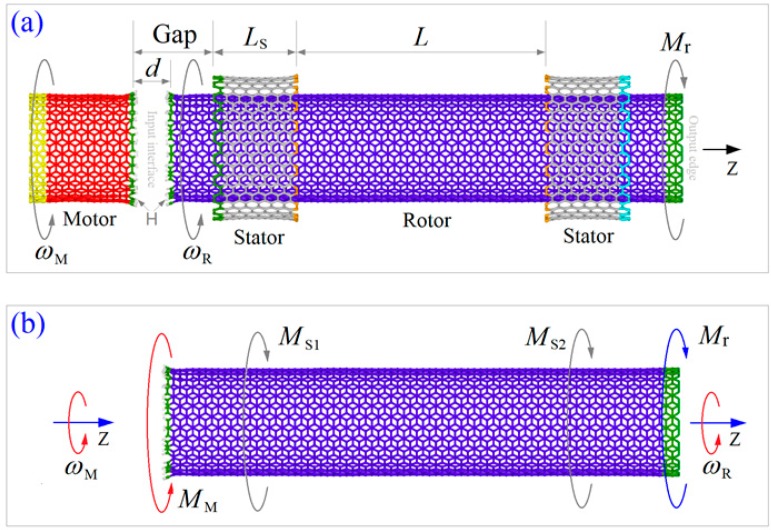
The initial model for testing the transmission torque moment from motor to rotor. (**a**) Configuration of the transmission model. The left short carbon nanotube acts as a motor with constant input rotational frequency of *ω*_M_. The right part is a nano-bearing from double walled carbon nanotubes (CNTs). The rotor is confined by the two same concentric stators. The rotational frequency of the rotor, i.e., *ω*_R_, is the output rotation. At the right edge of the rotor, an active resistant torque moment, *M*_r_, is applied for reducing the value of *ω*_R_. All the tubes are concentric with the same axis of Z. The essential geometrical factors are as Gap = 1.5 nm being the distance between the motor and the left stator. *L*_S_ is the length of a stator. *L* is the distance between the two stators. Axial commensurate CNTs are used to form an armchair model from armchair tubes (with C-C bonds as edges) or a zigzag model from zigzag tubes (with uniformly distributed atoms as edges). The neighboring edges of motor and rotor with an initial distance of *d* = 0.5 nm are hydrogenated. Detailed parameters are listed in Table 5. (**b**) Free body diagram of the rotor. *M*_M_ is the driving moment from the motor. *M*_S1_ and *M*_S2_ are from the two stators.

**Table 1 ijms-20-03851-t001:** Statistical results of *R*_Tran_ of the zigzag model and subjected to *M_r_* = 0 eV.

Temperature	*ω*_M_ = 50 GHz	*ω*_M_ = 100 GHz	*ω*_M_ = 150 GHz	*ω*_M_ = 200 GHz
T = 8 K	0.60 ± 0.028	0.74 ± 0.013	0.82 ± 0.009	1.0 ± 0.025
T = 50 K	0.54 ± 0.026	0.69 ± 0.048	0.80 ± 0.061	0.80 ± 0.061
T = 100 K	0.91 ± 0.180	0.68 ± 0.057	0.79 ± 0.084	0.64 ± 0.045
T = 300 K	0.95 ± 0.157	0.72 ± 0.082	0.77 ± 0.113	0.64 ± 0.081
T = 500 K	0.96 ± 0.150	0.70 ± 0.202	0.76 ± 0.122	0.11 ± 0.030

Mean value and standard deviation (Std), e.g., Mean ± Std, is given for each case.

**Table 2 ijms-20-03851-t002:** Critical values of the resistant moment, i.e., *M*_r_^cr^, on the rotor in the zigzag model. Unit of the moment: eV. Sign convention: Minus sign means the direction of *M*_r_ is opposite to that of *ω*_M_.

Temperature	*ω*_M_ = 50 GHz	*ω*_M_ = 100 GHz	*ω*_M_ = 150 GHz	*ω*_M_ = 200 GHz
T = 8 K	−3.0	−205(buckled)	−210(buckled)	−217.23(buckled)
T = 50 K	−0.393	−1.75	−175	−200(buckled)
T = 100 K	−0.32	−0.688	−0.625	−130
T = 300 K	−0.181	−0.338	−0.219	−0.5
T = 500K	−0.188	−0.25	−0.188	−0.3

**Table 3 ijms-20-03851-t003:** Statistical results, i.e., Mean ± Std, of *R*_Tran_ of the armchair model and subjected to *M*_r_ = 0 eV (see black lines in Figure 3).

Temperature	*ω*_M_ = 50 GHz	*ω*_M_ = 100 GHz	*ω*_M_ = 150 GHz	*ω*_M_ = 200 GHz
T = 8 K	0.97 ± 0.124	0.99 ± 0.049	0.89 ± 0.016	0.93 ± 0.009
T = 50 K	0.99 ± 0.089	0.71 ± 0.051	0.88 ± 0.071	0.67 ± 0.022
T = 100 K	0.99 ± 0.091	0.71 ± 0.056	0.81 ± 0.167	0.67 ± 0.043
T = 300 K	0.95 ± 0.138	0.68 ± 0.109	0.48 ± 0.051	0.37 ± 0.023
T = 500 K	0.53 ± 0.097	0.52 ± 0.095	0.40 ± 0.050	0.36 ± 0.033

**Table 4 ijms-20-03851-t004:** *M*_r_^cr^ on the rotor in the armchair model with Gap = 1.5 nm. Moment unit: eV.

Temperature	*ω*_M_ = 50 GHz	*ω*_M_ = 100 GHz	*ω*_M_ = 150 GHz	*ω*_M_ = 200 GHz
T = 8 K	−3.0	−250(buckled)	−260(buckled)	−275(buckled)
T = 50 K	−0.35	−0.75	−130	−250(buckled)
T = 100 K	−0.28	−0.469	−1.0	−100
T = 300 K	−0.125	−0.19	−0.25	−0.5
T = 500 K	−0.075	−0.175	−0.15	−0.25

**Table 5 ijms-20-03851-t005:** Parameters of the rotation transmission nanosystem with different models shown in Figure 4. Dimension unit: nm.

Model	*L*	Motor/Rotor				Stator/Stator			
		Chirality	Length	Diameter	Number of Atoms	Chirality	*L* _S_	Diameter	Number of Atoms
Zigzag	4.75	(26,0)	2.20/9.87	2.04	572C+26H/2444C+26H	(35,0)	1.56	2.74	560C/560C
Armchair	4.64	(15,15)	1.97/9.84	2.03	510C+30H/2430C+30H	(20,20)	1.60	2.71	560C/560C

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
