# Peer review of "Critical Output Torque of a GHz CNT-Based Rotation Transmission System Via Axial Interface Friction at Low Temperature"

_ijms, 2019, doi:10.3390/ijms20163851_

Round 1

Reviewer 1 Report

The manuscript presents a continuation of computational study of an artificial nanoscale motor based on double-wall carbon nanotubes (CNT) where rotation of the CNT inner-shell with respect to the stationary CNT outer-shell is driven by another CNT rotation. Such motor is called a rotation transmission system and operates at 50-200 GHz range. The authors analyzed the resistant torque effect using molecular dynamic simulations of the rotation transmission system at different temperature and drive frequency, and compared the effect on CNTs coupled by zig-zag and armchair edges. The presented results reproduce the effects discussed in published works, and give some additional knowledge on the system performance. It is interesting in the corresponding field for use in future molecular drivers. The manuscript is well prepared with 46 references, 4 figures and 4 tables. However, corrections have to be made to the presentation before publishing in your journal.

COMMENTS

Major

1. A model of the CNT-based system is given in Fig.1. However, the difference between armchair and zigzag edges is not described. It is better to illustrate it clearly for broader audience.

2. It is unclear whether the torque moments are vector quantity. In Eq (7) and line 134, you are using a vector product (x). This must be clearly stated.  

3. You are describing an idea to explain the temperature effect of the CNT expansion on line 191-200. However, no evidence is given. Please provide some evidences, for instance, the change of the mean “d”, i.e. the M-R distance, or the mean interface force, or the frequency of bonding interactions of the interface atom with temperature. Additionally, the different thermal expansion of CNT diameters for inner- and outer-shells have to be discussed as it brings the shells close and changes the Rotor-Stator coupling and therefore, the torque.

4. Because the CNTs are good electric conductors, the electrostatic capacitive coupling should be considered in the system as an additional friction force. I did not find any words about it. Please elaborate.

5. The title seems to be misleading. I would suggest “Critical torque of.. “. Also, please consider to include “carbon nanotubes” because the results are specific to this material.

Technical:

You are citing “0” in many places: line 46, 59, 89, 95….

Fig.3, the meaning of red, blue and green segments is not given. Also, the motor (red) is absent in 53ps and 56ps frames in part (c). Is the motor broken?

Line 92, the formatting of Table 1 is corrupted.

Some terms are very specific to the field. Such as “interface projects”, “Armchair model”, “generatrix”... Please elaborate for broader audience.

Author Response

Comments and responses

The manuscript presents a continuation of computational study of an artificial nanoscale motor based on double-wall carbon nanotubes (CNT) where rotation of the CNT inner-shell with respect to the stationary CNT outer-shell is driven by another CNT rotation. Such motor is called a rotation transmission system and operates at 50-200 GHz range. The authors analyzed the resistant torque effect using molecular dynamic simulations of the rotation transmission system at different temperature and drive frequency, and compared the effect on CNTs coupled by zig-zag and armchair edges. The presented results reproduce the effects discussed in published works, and give some additional knowledge on the system performance. It is interesting in the corresponding field for use in future molecular drivers. The manuscript is well prepared with 46 references, 4 figures and 4 tables. However, corrections have to be made to the presentation before publishing in your journal.

COMMENTS

Major

1. A model of the CNT-based system is given in Fig.1. However, the difference between armchair and zigzag edges is not described. It is better to illustrate it clearly for broader audience.

Re: Armchair tube has bond edges, i.e., C-C bonds at an edge are vertical to the tube axis. Zigzag tube has atom edges, i.e., the bonds formed by the edge atoms are not vertical to the tube axis, but 1/3 of internal C-C bonds are parallel to the tube axis. As CNTs have been widely studied, we just give their major parameters in Table 1.

2. It is unclear whether the torque moments are vector quantity. In Eq (7) and line 134, you are using a vector product (x). This must be clearly stated. 

Re: Because the vectors have the same direction, and only the axial components of either torque moments or rotational accelerations are considered, hence, vector operation is unnecessary here. For Eq.(7), we change the cross into a dot.

3. You are describing an idea to explain the temperature effect of the CNT expansion on line 191-200. However, no evidence is given. Please provide some evidences, for instance, the change of the mean “d”, i.e. the M-R distance, or the mean interface force, or the frequency of bonding interactions of the interface atom with temperature. Additionally, the different thermal expansion of CNT diameters for inner- and outer-shells have to be discussed as it brings the shells close and changes the Rotor-Stator coupling and therefore, the torque.

Re: If necessary, a new citation of article on thermal expansion of CNTs can be added. For example, “Jiang H, Liu B, Huang Y, Hwang KC. Thermal expansion of single wall carbon nanotubes. Journal of Engineering Materials and Technology, 2004, 126:265-270.” As labeled in the revised manuscript.

4. Because the CNTs are good electric conductors, the electrostatic capacitive coupling should be considered in the system as an additional friction force. I did not find any words about it. Please elaborate.

Re: The system keeps electric neutrality. Hence, the effect of electrostatic forces is not considered even though CNTs are good conductors. If electric neutrality is not kept, the potential energy function should be modified, e.g., Coulombian forces among atoms should be considered in MD simulation.

5. The title seems to be misleading. I would suggest “Critical torque of.. “. Also, please consider to include “carbon nanotubes” because the results are specific to this material.

Re: Modified accordingly.

Technical:

You are citing “0” in many places: line 46, 59, 89, 95….

Re: We are not sure the meaning of “citing 0”.

Fig.3, the meaning of red, blue and green segments is not given. Also, the motor (red) is absent in 53ps and 56ps frames in part (c). Is the motor broken?

Re: In Fig. 3, we set the atoms in stators to be in red or green, and the atoms in the rotor in blue or green. As the figure is perspective of the system from the axial direction, the near end of a tube looks thicker than its far end. The green atoms on both the rotor and the stators are used to show the relative rotational of the rotor with respect to the stator. It does not mean the motor is broken.

Line 92, the formatting of Table 1 is corrupted.

Re: Table 1 is modified.

Some terms are very specific to the field. Such as “interface projects”, “Armchair model”, “generatrix”... Please elaborate for broader audience.

Re: Conditionally modified.

Reviewer 2 Report

The paper by Wu et al. presidents a classical molecular dynamics study of a complex system comprising carbon nanotube bearings and rotary parts. The main conclusions refer to the role of friction in the transmission of the rotational motion. While the simulation setup appears to not be original, the findings are important enough to be publishable. My main concern is related to the L-J treatment of the inter-tube interactions which for example leads to a chirality independent effect. A recent work [Extreme Mechanics Letters, Volume 30, 2019, 100508] indeed show that a corrugation effect occurring via a KC treatment at the interfaces is indeed important. In a revised version, the authors should put their results in the context of these recent findings.  

Author Response

Re: Yes, L-J interaction depends on the relative positions of non-bonding atoms. However, chirality difference between armchair and zigzag CNTs leads to different potential energy barrier at the two types of edges. Hence, the magnitude of interaction between two tubes depends on both the chirality and their distance. This property has been demonstrated for more than a decade in many publications. Hence, we just cite a reference here, e.g., [46] Z. Guo, T. Chang, X. Guo and H. Gao, Thermal-induced edge barriers and forces in interlayer interaction of concentric carbon nanotubes, Phys. Rev. Lett., 2011, 107, 105502.